# Shock-Absorber Rotary Generator for Automotive Vibration Energy Harvesting

**Tae Dong Kim and Jin Ho Kim \***

Department of Mechanical Engineering, Yeungnam University, Gyeongsan 38541, Korea;
rlaxoehd3311@ynu.ac.kr

**\*** Correspondence: jinho@yun.ac.kr

**Abstract:** The vibration energy derived from vehicle movement over a road surface was first converted to rotational energy during vehicle operation by installing blades in the suspension system. The rotational energy was converted to electrical energy using the rotational energy as the input value of the rotary generator. The vibrations from the road's surface were analyzed using CarSim-Simulink. The blades' characteristics were analyzed using ANSYS Fluent. The T–ω curve was derived, and the power generation of the rotary generator was verified using the commercial electromagnetic analysis program, ANSYS MAXWELL. For high power generation, the design was optimized using PIAnO (process integration, automation, and optimization), a PIDO (process integration and design optimization) tool. The amount of power generation was 59.4562 W, which was a 122.47% increase compared to the initial model. The remaining problems were analyzed, and further studies were performed. This paper proposes the applicability and development direction of suspension with energy harvesting by installing blades on suspension.

**Keywords:** energy harvesting; vehicle suspension; rotary generator; CarSim-Simulink; optimal design; ANSYS Fluent; ANSYS MAXWELL

---

## 1. Introduction

Research on electric vehicles has been conducted actively in line with the development and diffusion of electric vehicles [1–5]. Accordingly, the demand for electricity is increasing, and research on charging electric vehicles is being conducted actively [6,7]. In this paper, energy harvesting was performed using energy harvesting technology. Energy harvesting technology is a technology that collects small amounts of energy that are discarded routinely or not used and converts this energy into usable electrical energy. This technology has been highlighted as a new and renewable energy source technology. Moreover, this is an eco-friendly energy utilization technology that can maintain the stability, security, and sustainability of the energy supply, because it obtains electric energy directly from nature and can reduce environmental pollution. Other researchers are also working on energy harvesting [8–10]. The vibration energy transmitted to the vehicle's body from the road's surface during operation can be recovered and converted to usable electrical energy to enable self-power generation functions.

The generators used can be divided into rotary generators and linear generators. When utilizing the energy harvesting function as a linear generator, the power generation results are somewhat less efficient as a result of the "end-effect". Figure 1 presents power generation through a linear generator [11].

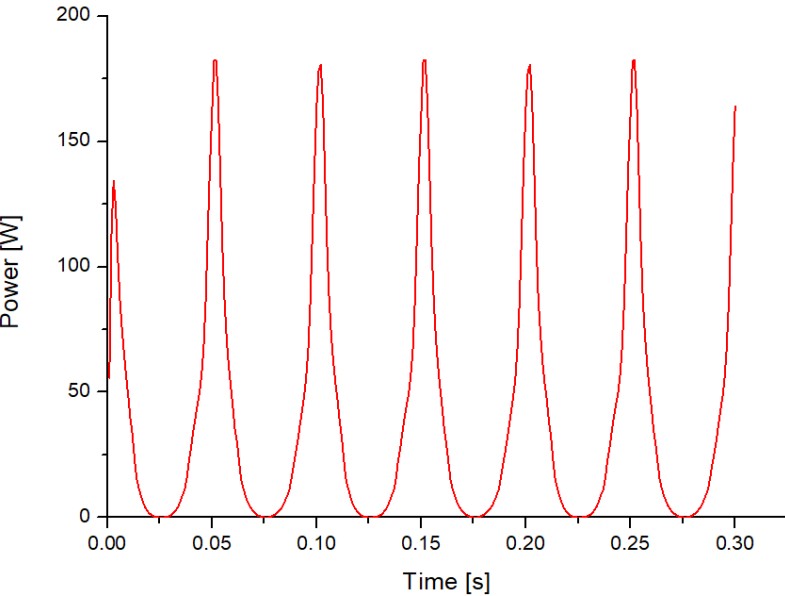

**Figure 1.** End-effect.

In addition, linear generators are complicated to manufacture along the exact center. As a result, in this study, the results of the power generation were verified using a rotary generator.

The vibration energy from the road's surface was analyzed using CarSim-Simulink to calculate the flow velocity in the suspension system. After removing the orifice in the suspension system, the blade was installed and designed to convert the rotational energy to vibration energy. Computational fluid dynamics (CFD) verified the characteristics of the blades. The resulting rotational energy was converted to electrical energy using a rotary generator, and the amount generated was examined using ANSYS MAXWELL. To achieve high power generation, the design was optimized using an orthogonal array provided by PIAnO (process integration, automation, and Optimization), a PIDO (process Integration and design optimization) tool. This study assessed the installation of the blades on the suspension system to determine the applicability and development direction of the suspension system with an energy harvesting function.

## 2. Structure of Electromagnetic Suspension

The suspension system consisted of a shock absorber, a spring, and a suspension arm. This system performs various functions such as ensuring the stability of vehicle steering, providing a comfortable ride, maintaining the proper height of the vehicle, lowering the impact, maintaining wheel alignment, weight support of the vehicle, and maintaining the tire tread status. The conventional device is a viscous hydraulic system used to suppress and dissipate vibrations by the pressure difference as the fluid passes through the orifice. As mentioned earlier, in this study, the blade was installed in the shock absorber, and the blade suppressed the vibrations instead of the orifice. Figure 2 presents the half model of the present device used in this study.

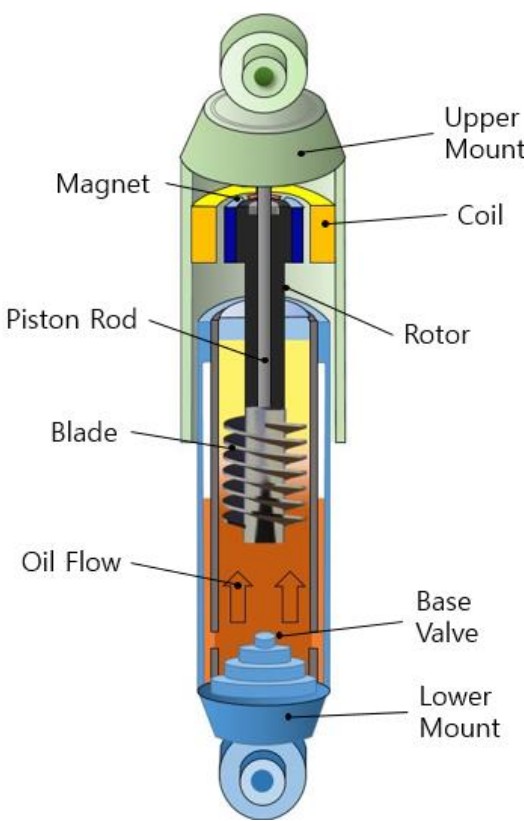

**Figure 2.** Cross-section of the suspension model used in this study.

Vibration energy from the road's surface (i.e., up and down motion) becomes an upward and downward motion of a fluid, and the flow of the fluid is converted to the rotation of the blade. The piston only moves up and down; it does not rotate. Rotors can be rotated because of the presence of an air gap and bearings, shown in Figure 3a. (Figure 3b provides the A–A section view).

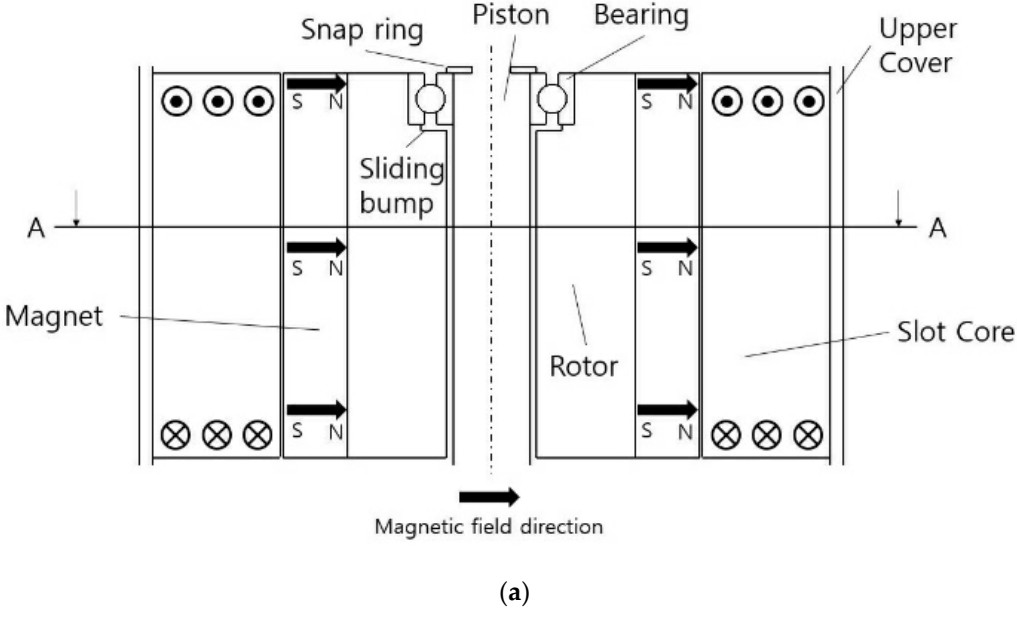

(a)

**Figure 3.** *Cont.*

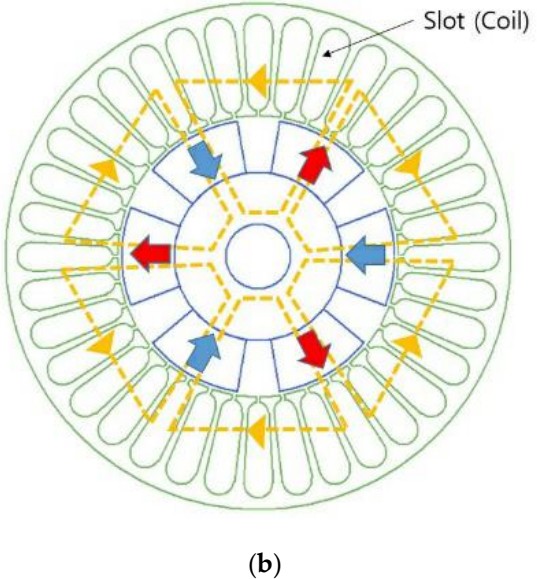

**(b)**

**Figure 3.** Structure explanation: (**a**) electromagnetic suspension structure; (**b**) A–A section view.

An air gap exists between the rotor and the piston, and they maintain non-contact. The bearings allow the rotor to rotate. The piston moves up and down with the rotor and installs a snap ring so that it does not move separately from the piston of the rotor. The snap ring also holds the inner wheel of the bearing, and the sliding bump of the rotor holds the outer wheel of the bearing. The outer and inner wheels of the bearing determine the dimensions of the snap ring and the sliding bump, respectively. Figure 3b shows the shape of the rotary generator. A PM (permanent magnet) is attached to the outer wall of the rotor, and there is a slot core on the inner wall of the upper cover. The rotor rotates together with a magnet while rotating. Relative movement with a coil is formed, and electric energy is generated because of a change in the magnetic flux with time. Electrical energy follows Faraday's Law, and the equation can be expressed as follows:

$$e(t) \ = \ -N\frac{d\varnothing}{d\theta}\frac{d\theta}{dt} \tag{1}$$

In this case, $N$ is the number of turns of the coil; $\varnothing$ is the flux passing through each turn during time $t$, and $\frac{d\theta}{dt}$ is the rotational speed of the mover. Figure 4 is the schematic diagram, which shows the overall procedure of the paper.

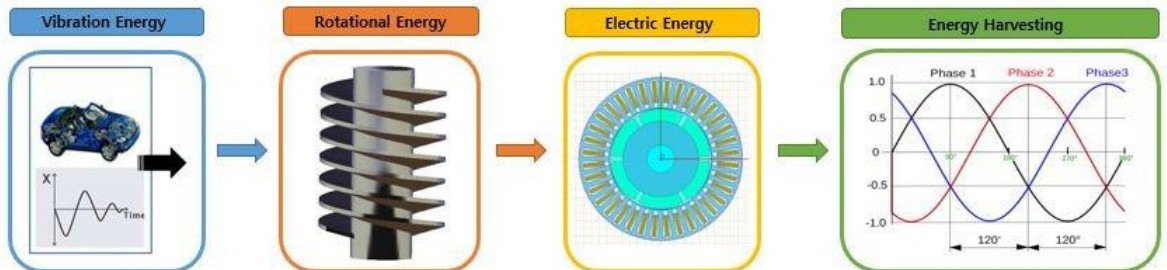

**Figure 4.** Schematic diagram.

## 3. Vibration Analysis with CarSim-Simulink

CarSim is a software that simulates a variety of vehicles to analyze the response of the vehicle. Simulink, which exists within MATLAB, is a model-based design tool made from the block diagram of the system. Several simulations and vehicle studies have been conducted, including driving

simulations and vehicle dynamics simulations, using both CarSim and Simulink. Recently, in 2018, a study of a cruise control system based on fuzzy PID control was conducted to assist the driver [12].

Figure 5 presents the vehicle model, and Table 1 lists the main characteristics of the vehicle.

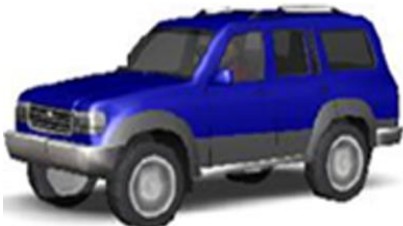

**Figure 5.** Vehicle model.

**Table 1.** Vehicle model characteristics.

| Parameters | Value |
| --- | --- |
| Sprung mass | 2257 kg |
| Unsprung mass | 100 kg |
| Wheelbase | 2946 mm |
| Internal engine model | 250 kW |
| Speed of vehicle | 80 km·h$^{-1}$ |

For the vehicle simulation test, a block diagram for the vehicle driving simulation control shown in Figure 6 was designed using MATLAB Simulink [13]. As the driving simulation test output variables, the displacement of the road's surface meeting the tire, the vibration displacement of the vehicle body and the wheel, the displacement of the suspension and the damping force of the suspension were set. Figure 7 shows the roughness of the road, and the speed of the included suspension of each wheel was derived. After calculating the speed of the four suspensions with the RMS (root mean square), the mean value was calculated to be 0.2138 m·s$^{-1}$.

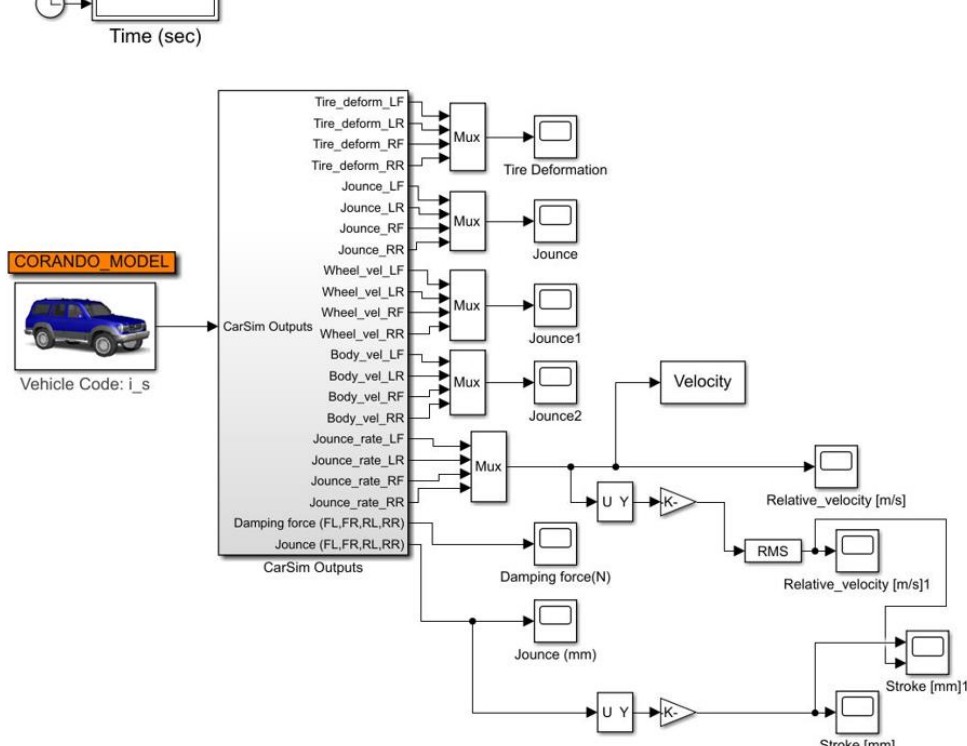

**Figure 6.** Block diagram.

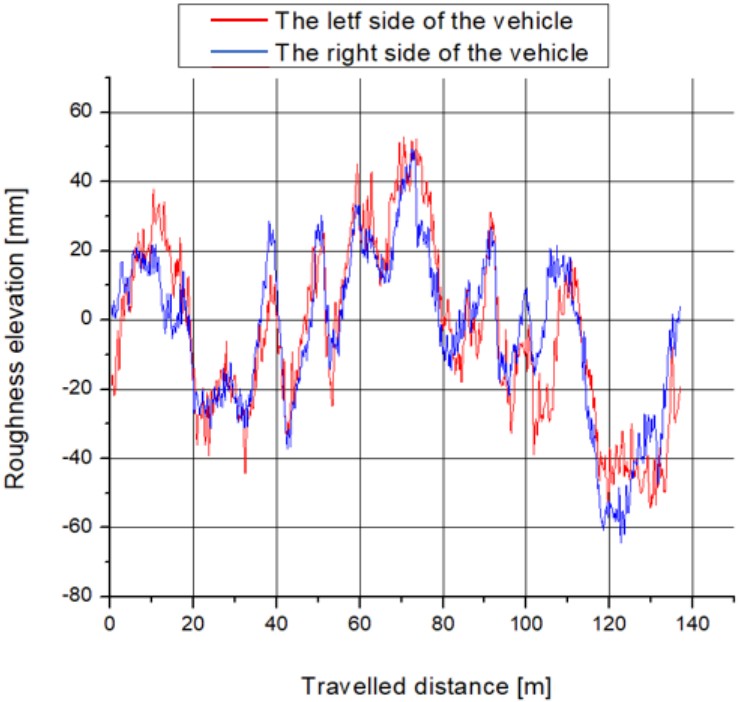

**Figure 7.** Road roughness.

## 4. Analysis of the Blade Characteristics with ANSYS Fluent

### 4.1. Pro-Processing Step

The characteristics of the blades by fluid flow delivered from the road's surface were analyzed using ANSYS Fluent. The main aim of this analysis was to derive the A–A curve of the blade. Figure 8 shows the modeling of the flow analysis.

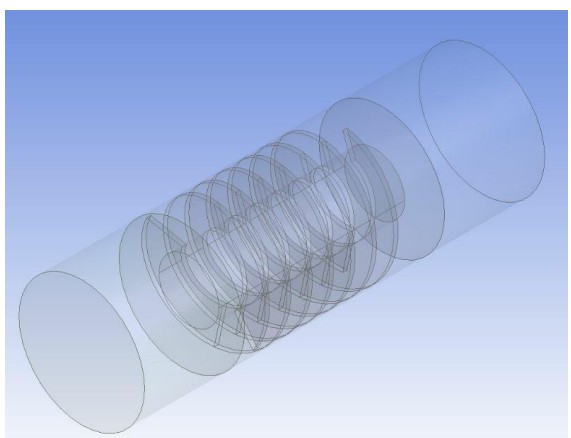

**Figure 8.** ANSYS Fluent modeling.

The blades were modelled using commercial software CATIA. Using the Import function, the blades were produced in ANSYS Geometry, and the fluid around the blades was modeled using the Subtract function in Boolean.

The oil SAE10W~SAE30W oil is used as the shock absorber [14]. The fluid used in this analysis was SAE 30 W. Figure 9 and Table 2 present the characteristics of the fluid according to the change in temperature.

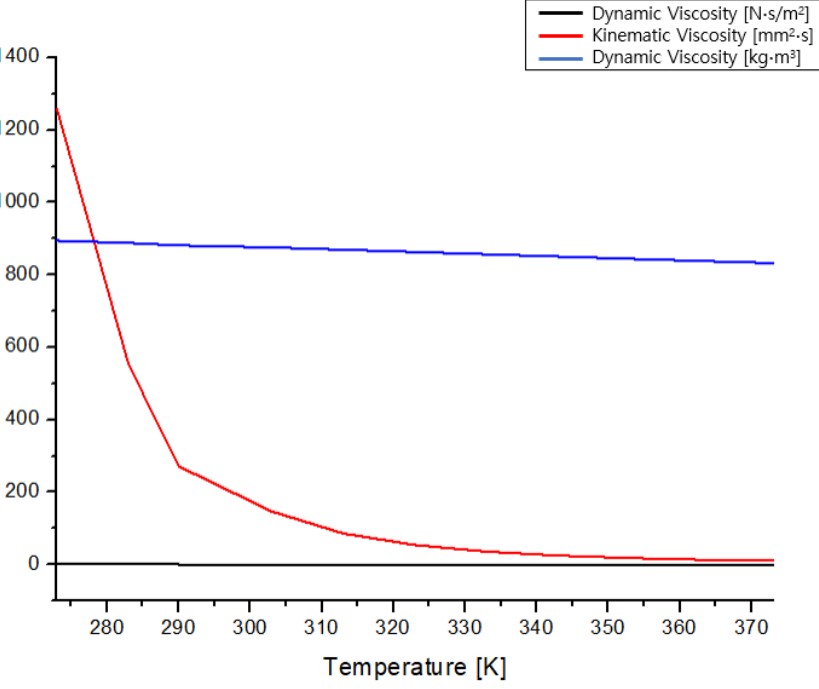

**Figure 9.** Characteristic graph of the fluid.

**Table 2.** Characteristics of the fluid at normal temperature.

| Temperature (K) | Dynamic Viscosity (N·s/m$^2$) | Kinematic Viscosity (mm$^2$·s$^{-1}$) | Density (kg·m$^{-3}$) |
|---|---|---|---|
| 273 | 1.1241 | 1257.25 | 894.1 |
| 283 | 0.4911 | 553.2 | 887.8 |
| 290 | 0.23939 | 271.56 | 881.5 |
| 303 | 0.12842 | 146.7 | 875.4 |
| 313 | 0.07455 | 85.76 | 869.3 |
| 323 | 0.04643 | 53.8 | 863.0 |
| 333 | 0.03058 | 35.69 | 856.9 |
| 343 | 0.02117 | 24.89 | 850.6 |
| 353 | 0.01528 | 18.1 | 844.4 |
| 363 | 0.01142 | 13.62 | 838.3 |
| 373 | 0.0088 | 10.58 | 832.2 |

For simplicity of analysis, the fluid temperature was chosen as the normal temperature. Table 3 lists the fluid properties at normal temperature.

**Table 3.** Characteristics of the fluid in this study.

| Parameters | Value |
|---|---|
| Temperature | 293 K |
| Dynamic Viscosity | 0.2394 N·s/m$^2$ |
| Kinematic Viscosity | 271.56 mm$^2$·s$^{-1}$ |
| Density | 881.5 kg·m$^{-3}$ |

The Reynolds number was calculated to select a model for CFD. This checked whether it was laminar or turbulent, and the equation is expressed as follows:

$$Re = \frac{inertia\ force}{viscous\ force} = \frac{\rho VD}{\mu} = \frac{VD}{\nu} = \frac{881.5 \times 0.2138 \times 0.09}{0.2394} = 70.8514 \tag{2}$$

where $V$ = average velocity of the flow; $D$ = characteristic length; $\mu$ = dynamic viscosity; $\nu$ = kinematic Viscosity; $\rho$ = density.

The Reynolds number was calculated to be 70.8514, indicating that the model in this analysis proceeded in laminar flow. The blade was rotated using the Reference Frame and Mesh Motion functions in ANSYS Fluent. Mesh Motion moves the mesh to follow the changing geometry of the boundary by moving the points at all stages of the analysis. Figure 10 shows the mesh used in the analysis.

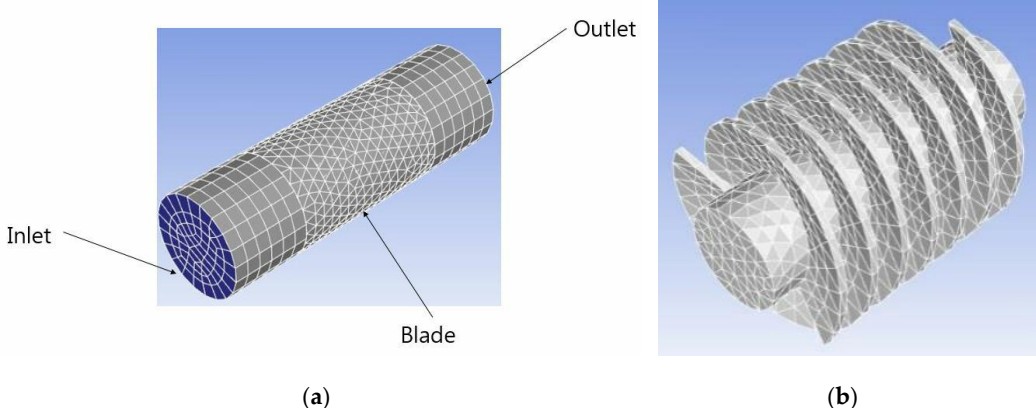

(**a**)　　　　　　　　　　　　　　　　(**b**)

**Figure 10.** CFD Mesh: (**a**) mesh; (**b**) blade mesh.

The input data were the inlet velocity of the fluid and the angular velocity of the blade. The velocity of the inlet calculated using CarSim-Simulink was 0.2138 m·s$^{-1}$. The torque was measured by changing the angular velocity of the blade. Zero rad/s was used as the initial value, and the angular velocity was increased gradually so that the resulting torque was 0.

*4.2. Post-Processing Step*

As mentioned above, the torque according to the angular velocity was calculated to derive the T–$\omega$ curve. The time step of the analysis started at 0.0001 s and decreased to 0.00005 s as the angular velocity increased. A different time step was set because the angular velocity increased; a sizeable cumulative error may occur from the previous time step to the next time step without calculation. The errors were calculated repeatedly until they were less than 10$^{-4}$. The analysis was conducted at 1 rad/s intervals, and the torque was negative at 28 rad/s. Figure 11 shows the derived T–$\omega$ curve [15].

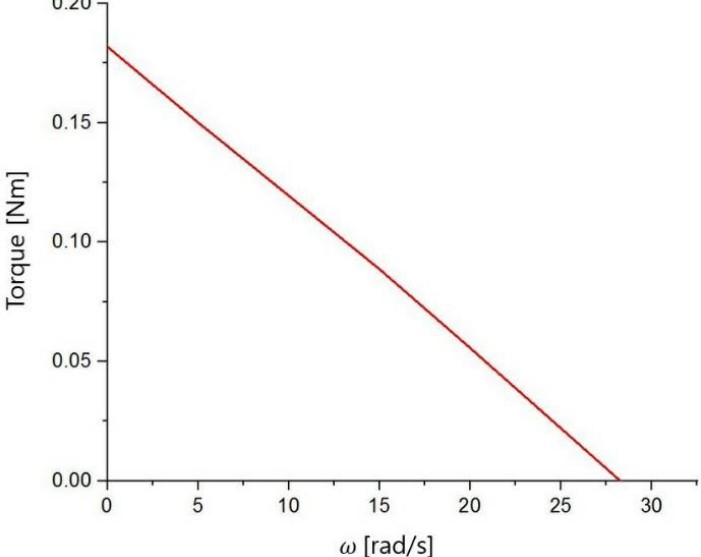

**Figure 11.** T–$\omega$ curve.

Figure 12 shows the pressure applied to the blade at $\omega$ = zero, 14, and 28 rad/s. The pressure was expressed as the sum of the viscosity and torque, confirming that the pressure applied to the blade decreases with increasing $\omega$.

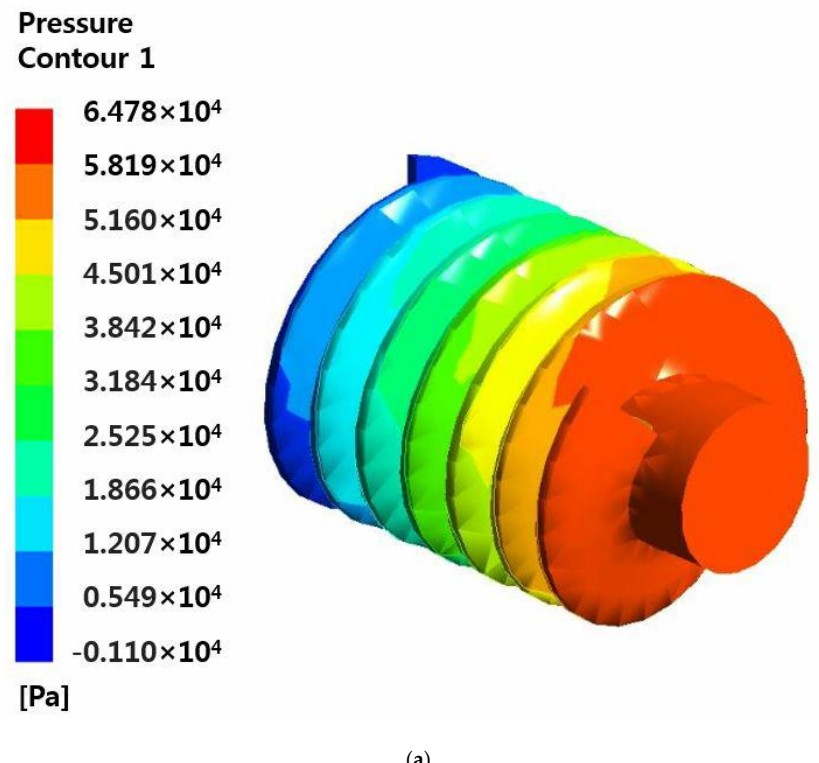

(**a**)

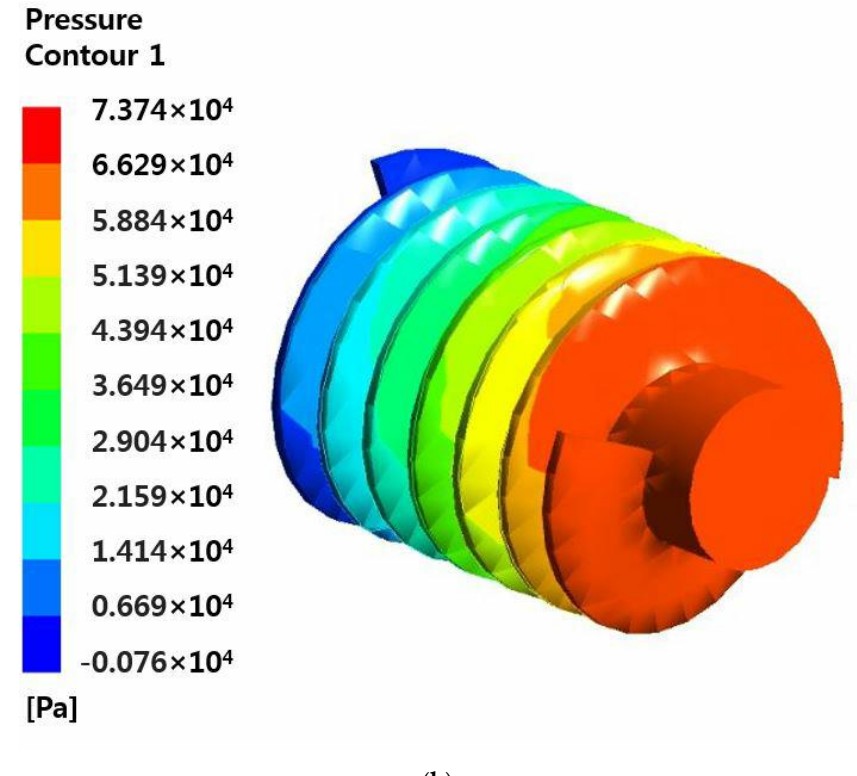

(**b**)

**Figure 12.** *Cont.*

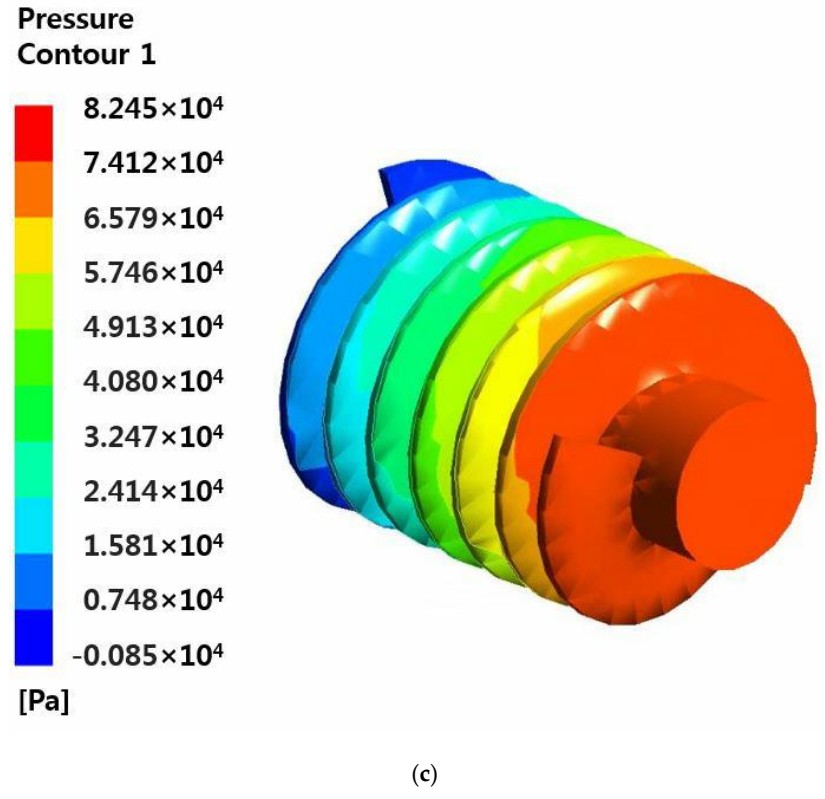

(**c**)

**Figure 12.** Pressure on the blade: (**a**) $\omega$ = 0 rad/s; (**b**) $\omega$ = 14 rad/s; (**c**) $\omega$ = 28 rad/s.

In this study, the value of $T \times \omega$ was selected to be the maximum value to verify the maximum generation of the rotary generator. The calculation can be made using Equation (2), and $\omega$ was chosen as 14 rad/s.

$$P_m = T \times \omega = V \times I = P_E, \tag{3}$$

In the process of energy conversion, it was assumed that there was no copper loss, iron loss, or machine loss.

## 5. Analysis of Rotary Generator with ANSYS MAXWELL

The generator was simulated using the commercial electromagnetic analysis program, ANSYS MAXWELL. RMxprt is a software convenient for modelling spinning machines and uses RMxprt to model rotary generators. The upper cover and rotor of the suspension system determine the inner and outer diameters of the rotary generator, and the air gap was set to 0.8 mm. Table 4 lists the main parameters.

**Table 4.** Major Parameters of the Rotary Generator.

| Parameters | Unit | Quantities |
|---|---|---|
| Number of Pole | | 6 |
| Number of Phases | | 3 |
| Number of Slot | | 36 |
| Outer Diameter of Rotor | mm | 35.4 |
| Outer Diameter of Stator | mm | 106 |
| Magnet Thickness | mm | 11 |
| Embrace | | 0.7 |

Figure 13 presents the modeling performed at RMxprt. The Rotor Core selected S20C material to flow the magnetic flux well, and Nd–Fe with a high magnetic force was used as a permanent magnet.

The diameter of the wire was set to 0.541 mm, and the "fill factor" determined the ratio at which the conductor occupied the cross-sectional area of the slot [16].

$$\lambda_s = \frac{Winding\ Area}{Total\ Slot\ Area} = 0.75,\tag{4}$$

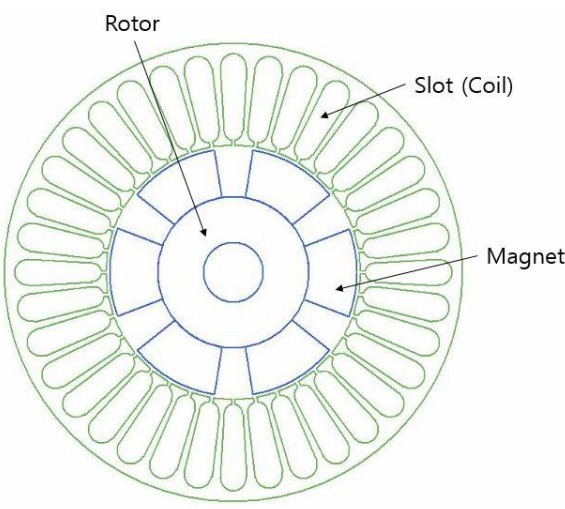

**Figure 13.** A 6 pole, 3 phase, 36 slot generator.

After RMxprt completed the analysis, it was designed as 2D MAWELL. Figure 14 shows the modeling and mesh of 2D MAXWELL.

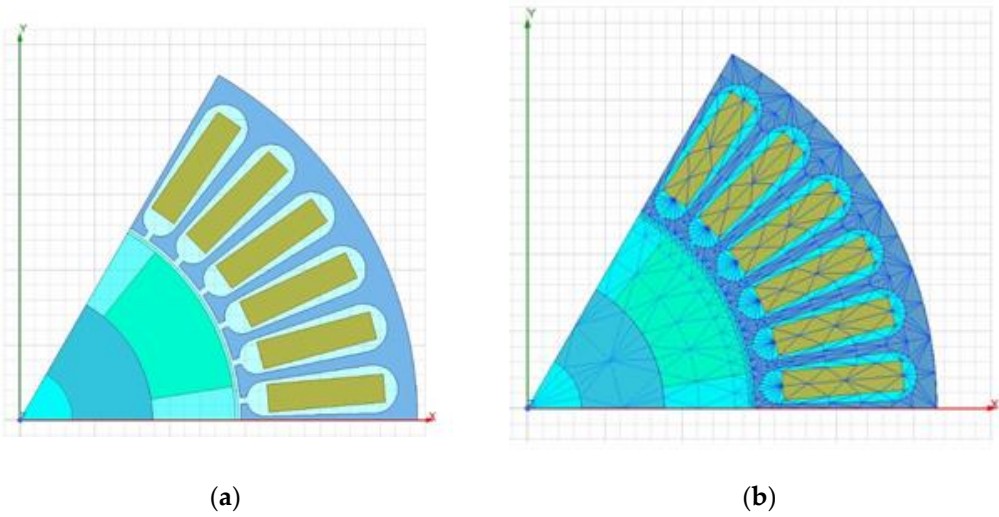

(**a**)　　　　　　　　　　　　　　　　　　　　　　　(**b**)

**Figure 14.** 2D MAXWELL Modeling: (**a**) modeling; (**b**) mesh.

Considering the analysis time, the simulation was analyzed as a 1/3 model and proceeded to a stop time of 0.45 s and a time step of 0.0001 s.

Figure 15 shows the results of generator analysis. The average power generation was 48.5452 W.

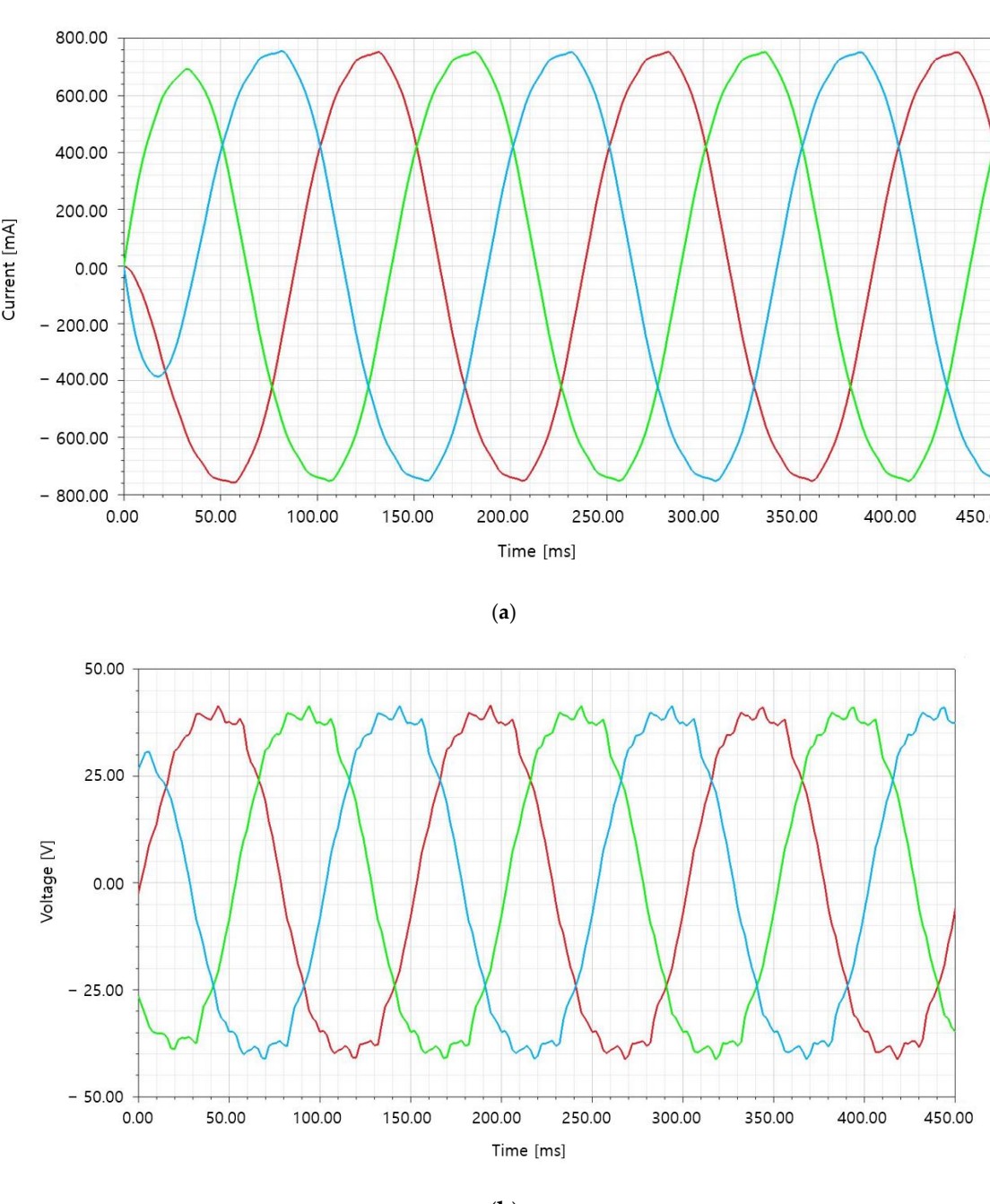

**(a)**

**(b)**

**Figure 15.** Initial model simulation results: (**a**) current of initial model; (**b**) voltage of initial model.

## 6. Rotary Generator Optimal Design

### 6.1. Design Variables and Constraints

In this study, the optimal design was carried out by adjusting the variables for maximum power generation. This is because the amount of power generation varies according to the size and position of the magnet, coil, and core of the generator. Three design variables were selected in four levels for effective design, and Figure 16 shows the selected variables. Stator $D_i$, magnet thickness, and embrace were selected to have a significant influence on power generation. The other dimensions were determined by the constraints, which were the inner diameter of the upper cover (106 mm), the piston diameter (14 mm), and the air gap (0.8 mm). Table 5 lists the range of each design variable, and the

desired result from the optimal design was the maximum power generation. As the Stator $D_i$ decreases or increases, the slot size also changes, and the number of turns of the coil also changes. The slot changes linearly according to the stator $D_i$ and the number of turns was set so that the coil had a fill factor of 0.75. Embrace has a maximum value of "1" and is a dimensionless number. This is defined as the ratio of the pole arc to pole pitch, and "1" means no space between two magnets [17].

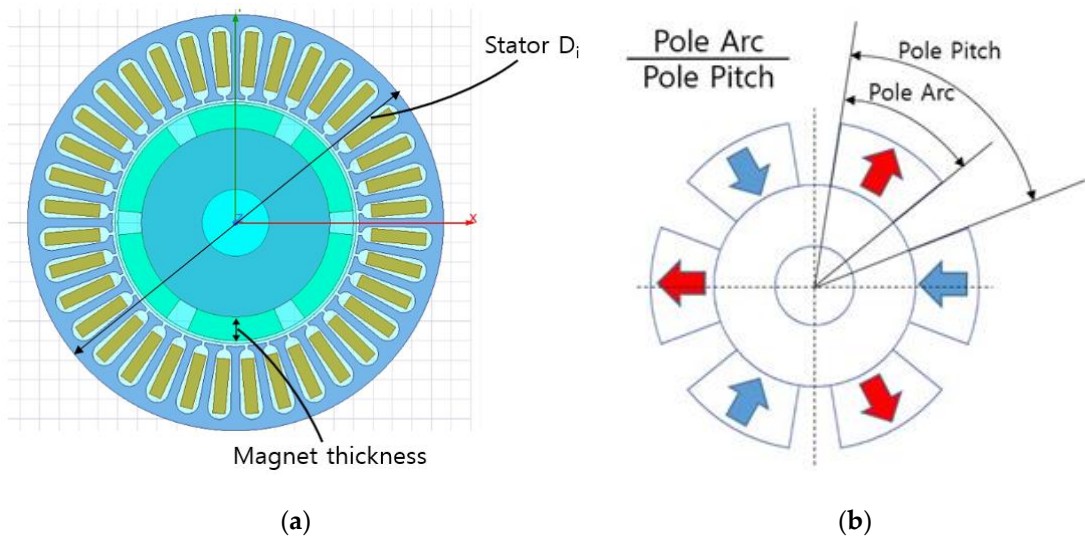

(**a**) (**b**)

**Figure 16.** Design variables of the optimal design: (**a**) Stator $D_i$, magnet thickness; (**b**) embrace.

**Table 5.** Boundary condition of the optimal design variables.

|  | Stator $D_i$ | Magnet Thickness | Embrace |
|---|---|---|---|
| Low | 59 mm | 9 mm | 0.7 |
| High | 62 mm | 12 mm | 1 |

### 6.2. Design of Experiment

For high power generation, 32 experimental points were obtained using an orthogonal array provided by PIAnO, a commercial process integration and design optimization (PIDO) tool. The power generation was calculated using the commercial electromagnetic analysis program, ANSYS MAXWELL.

### 6.3. Approximation Technique

For the approximate model, the Kriging model was selected. Kriging, a representative interpolation model, shows excellent predictive performance in nonlinear systems with many design variables and is a statistical technique that predicts the surrounding values in a linear combination [18]. The simulation analysis results for the experimental points produced a Kriging model using the approximate techniques provided by the PIAnO program [19].

### 6.4. Optimization Technique

The optimal design was carried out using the Evolution Algorithm (EA) among the optimization techniques using an approximate model [20]. An evolutionary algorithm is a probabilistic optimal technique. The next population was generated by producing probability variables within a specific range from the set design variable, the parent population. Through the parent object and the next object group, the variables close to the desired design goals are selected, and the design variables are reconstructed.

The approximation technique confirmed the accuracy of the Kriging model's performance. The evolutionary algorithm produced the generator model using the variable values shown in Table 5.

Table 6 lists the initial model and the optimally designed generator dimensions, and Figure 17 presents the optimized generator.

**Table 6.** Initial model and optimization model.

|         | Stator $D_i$ | Magnet Thickness | Embrace |
|---------|--------------|------------------|---------|
| Initial | 59 mm        | 11 mm            | 0.7     |
| Optimal | 61 mm        | 9 mm             | 1       |

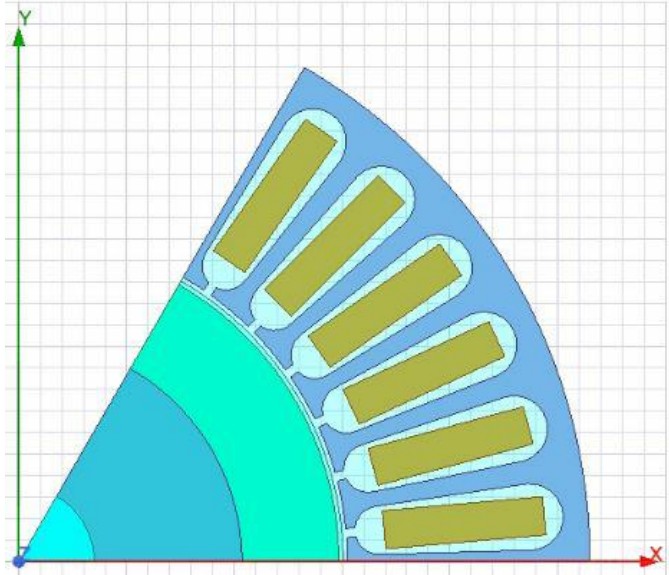

**Figure 17.** Optimal model.

### 6.5. Verification of Design Results

The optimal value was derived from the experimental point generated and verified that the power generation increased by 122.47% compared to the initial model, and it is shown in Table 7. The S/N Ratio (Signal-to-Noise Ratio) was checked to determine which variable is sensitive to the variable. Calculated according to Equation (5).

$$\eta = -10 \, log \, log \left( \frac{1}{n} \sum_{i=1}^{n} \frac{1}{y_i^2} \right) \tag{5}$$

**Table 7.** Power amount of the initial model and optimal model.

|         | P         | Increase Rate |
|---------|-----------|---------------|
| Initial | 48.5452 W | -             |
| Optimal | 59.4562 W | 122.47%       |

Figure 18 shows the S/N ratio of design variables, and Figure 19 presents the current, voltage graph, and flux of the generator generated through the optimal design.

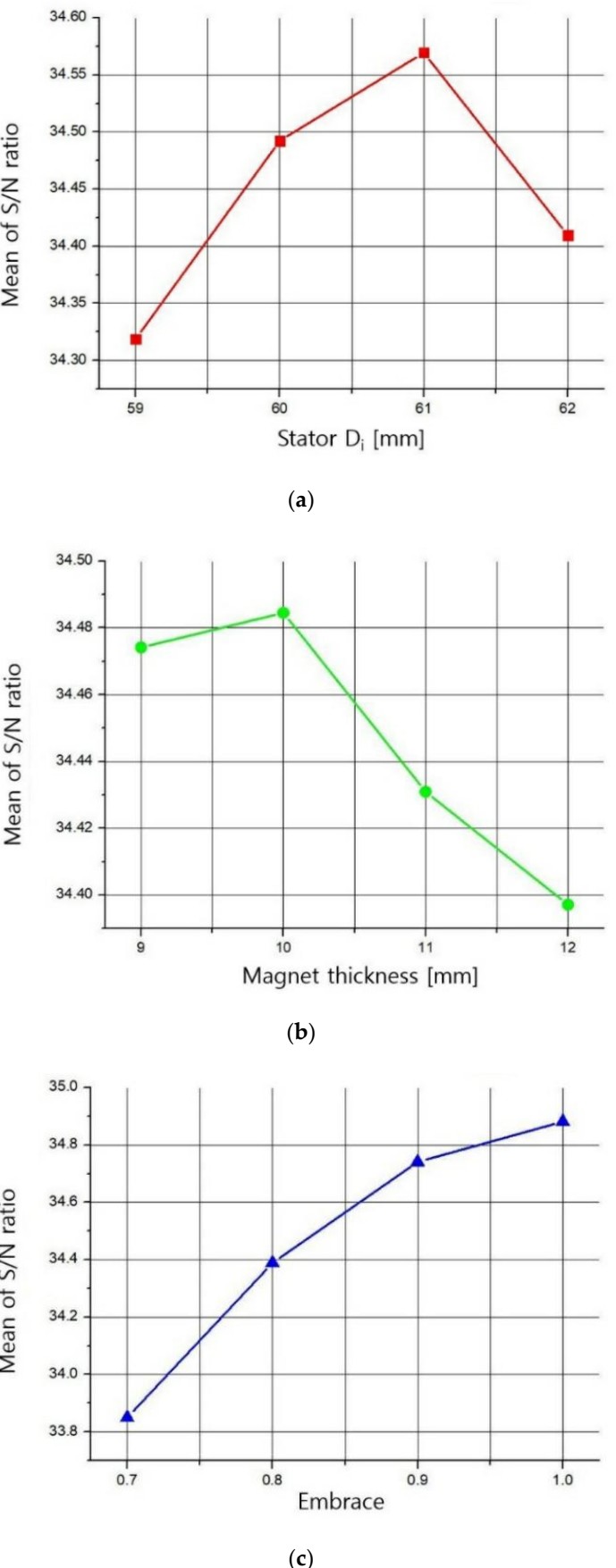

**Figure 18.** S/N ratio: (**a**) Stator $D_i$; (**b**) magnet thickness; (**c**) embrace.

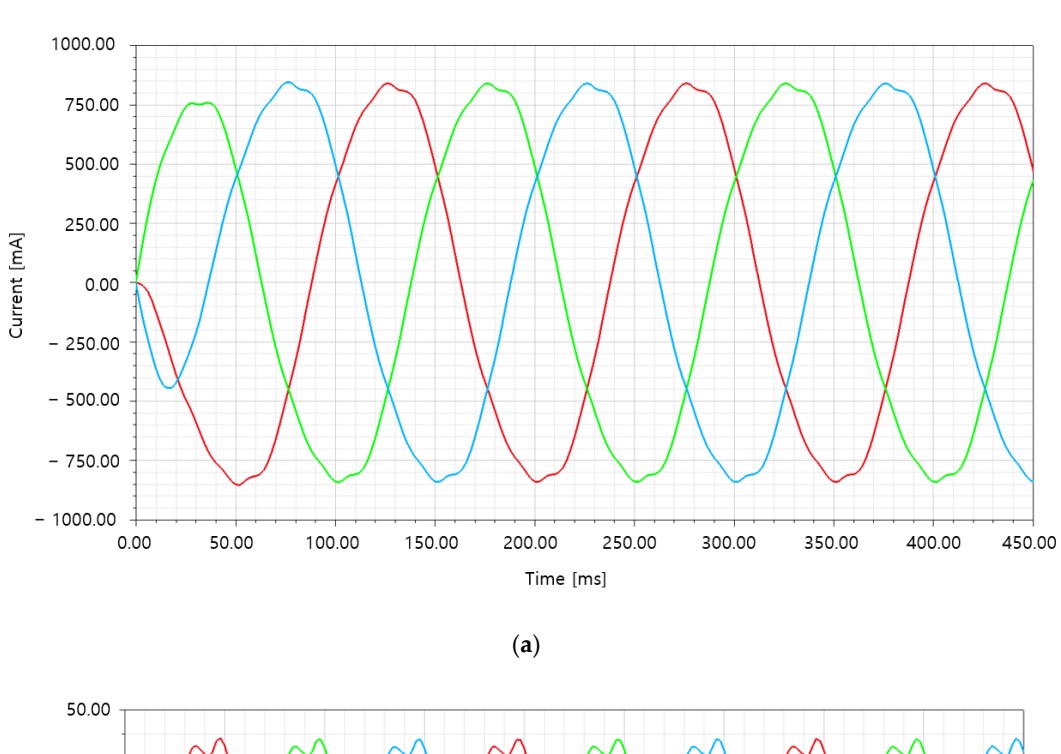

(**a**)

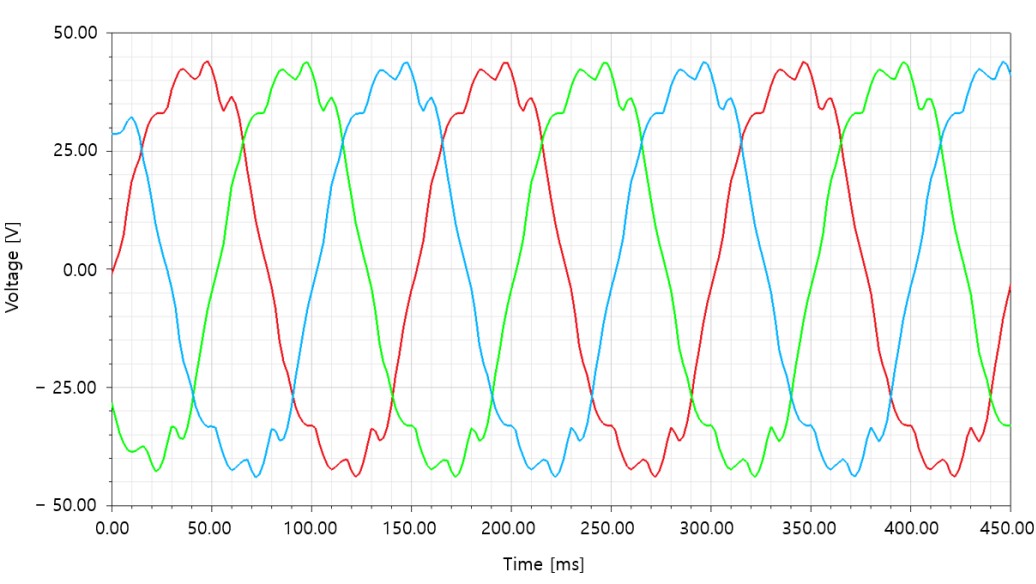

(**b**)

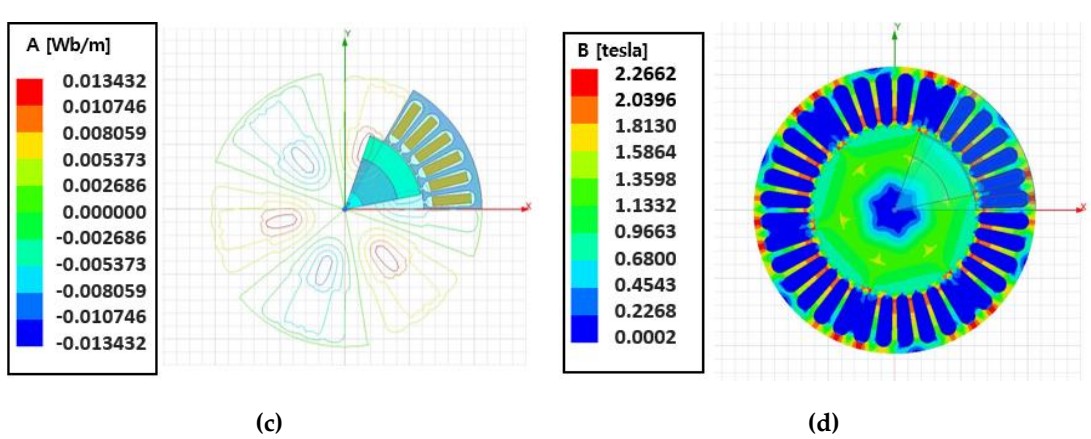

(**c**)
(**d**)

**Figure 19.** *Cont.*

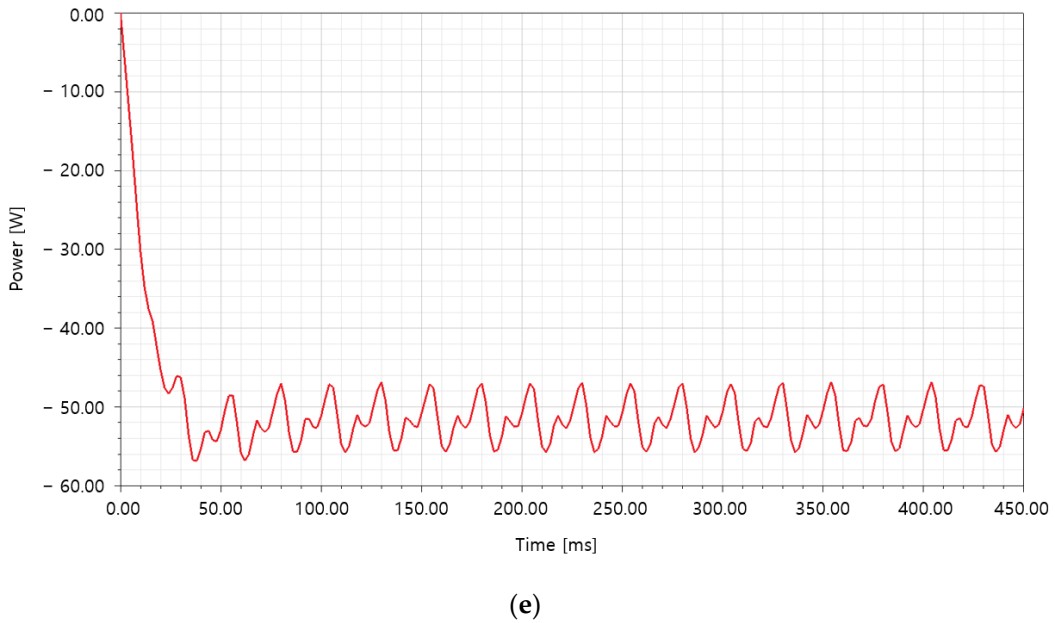

(**e**)

**Figure 19.** Optimal design results: (**a**) current of optimal model; (**b**) voltage of optimal model; (**c**) magnetic flux line; (**d**) magnetic flux density; (**e**) power.

## 7. Conclusions

This paper proposed the applicability and development direction of suspension with energy harvesting by installing blades on suspension. Blades were installed instead of the orifice, and a shock absorber was modeled including a rotor and piston, snap ring, and bearing. The vehicle's driving conditions were set using the CarSim-Simulink. The vibration characteristics were analyzed. The suspension speed was calculated, and the curve was derived using ANSYS Fluent. For maximum power generation, w was selected, and the selected w was interpreted as the input of the designed rotary generator. As a result, a value of 48.5452 W could be confirmed.

To increase the power generation significantly, the generator was optimized using an orthogonal array provided by PIAnO. As a result, 59.4562 W was generated, which is an increase of 122.47% compared to the initial model.

1. In this study, the blade was installed in place of the orifice of the existing suspension. Therefore, it necessary to check whether the installed blade generates as much damping force as the existing orifice.
2. The optimal design of the blades is needed to achieve the maximum power generation under limited conditions.
3. The addition of the rotors increased the load significantly, but further analysis and verification related are required.
4. The amount of power generation needs to be verified by inputting the velocity that changes over time instead of the average value of the inlet velocity of the fluid.
5. For simplified analysis, the fluid is marked at ambient temperature. Therefore, it will be necessary to calculate the energy equation by entering a change in the fluid temperature.
6. In the future, an analysis will be performed by changing the rotational speed of the generator, and the operation point will be identified by generating the generator $T$–$\omega$ curve.

In this study, many simplifications were made. Hence, the design will need to be optimized by reducing the simplifications in later studies.

**Author Contributions:** Conceptualization, J.H.K. and T.D.K.; software, T.D.K.; writing—original draft preparation, T.D.K.; writing—review and editing, J.H.K.; and T.D.K.; supervision, J.H.K.; funding acquisition, J.H.K. All authors have read and agreed to the published version of the manuscript.

**Funding:** This research was funding by Yeungnam University Research Grant in 2020.

**Acknowledgments:** This research was supported by Yeungnam University Research Grant in 2020.

**Conflicts of Interest:** The funders had no role in the design of the study; in the collection, analyses, or interpretation of data; in the writing of the manuscript, or in the decision to publish the results.

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
