# Peer review of "Shock-Absorber Rotary Generator for Automotive Vibration Energy Harvesting"

_applsci, doi:10.3390/app10186599_

Round 1

Reviewer 1 Report

See the attached review file

Author Response

Corrected. please check.

Reviewer 2 Report

A well structured paper. The results are supported with theory and simulation results.

1. In Fig. 18, please add labels for y aixs.

2. Authors also give simulation results on induced voltage. However, for an energy harvesting design, the actual power (in W) is more interesting to readers.

3. Please clarify how to extract the generated energy.

4. It's good to add a brief comparison table to compare your design with prior works.

Author Response

Corrected. please check.

Round 2

Reviewer 1 Report

The authors have addressed appropriately most of the comments/suggestions on the original manuscript, so the paper can be accepted for publication in its current form.

Author Response

Thank you for being a reviewer.
